# Nonparametric Functional Data Analysis for Forecasting Container Throughput: The Case of Shanghai Port

**Yuye Zou \*, Bohan Su and Yanhui Chen** 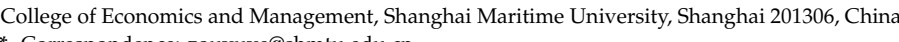

College of Economics and Management, Shanghai Maritime University, Shanghai 201306, China
\* Correspondence: zouyuye@shmtu.edu.cn

**Abstract:** Transportation is one of the major carbon sources in China. Container throughput is one of the main influencing factors of ports' carbon emission budget, and accurate prediction of container throughput is of great significance to the study of carbon emissions. Time series methods are key techniques and frequently used for container throughput. However, the existing time series methods treat container throughput data as discrete points and ignore the functional characteristics of the data. There has recently been interest in developing new statistical methods to predict time series by taking into account a continuous set of past values as predictors. In addition, to eliminate the linear constraint in the functional time series prediction approach, we propose a functional version of a nonparametric model that allows using a continuous path in the past to predict future values of the process, including functional nonparametric regression and functional conditional quantile and functional conditional mode models, to forecast the container throughput of Shanghai Port. For the purpose better forecasting, an experiment was conducted to compare our functional data analysis approaches with other forecasting methods. The results indicated that nonparametric functional forecasting methods exhibit more significant performance than other classical models, including the functional linear regression model, nonparametric regression model, and autoregressive integrated moving-average model. At the same time, we also compared the prediction accuracy of the three nonparametric functional methods.

**Keywords:** container throughput forecasting; functional data analysis; time series; nonparametric models

## 1. Introduction

The world economy has entered an important period of rapid development and globalization. As an important transportation hub connecting the sea and land, ports are a key link for each country to realize global international trade and construct a global logistics transportation system, international logistics, and supply chains. In view of the current situation of global economic development, ports play an important role in the development of the national economy. The operation of the port transport industry is a barometer of the country's macroeconomy, and the fluctuation of port throughput reflects the development of world trade. Due to the high efficiency of container handling, fast turnover, packaging cost savings, and cargo loss reduction, the container-based transportation system has gradually become the main way of maritime transportation. The container is an important index to evaluate the comprehensive capability of a port. Accurate prediction of container throughput is of great significance to port planning, operation, and decision-making and regional economic development. On the other hand, container throughput is one of the important factors affecting port carbon emissions, and accurate prediction of container throughput plays an important role in studying port carbon emissions.

There are many scholars using a variety of different methods to predict container throughput. The commonly used prediction methods include traditional models (multiple regression, exponential smoothing, grey prediction, neural networks, combinatorial and autoregressive models, etc.) and machine learning techniques (support vector machines,

random forests, deep learning algorithms, etc.). Although these methods have achieved certain predictive effects, they also have some limitations. For example, multiple linear regression models explore the linear relationship between variables based on fixed mathematical models. However, in many practical fields, especially in the economic field, there is often a nonlinear relationship between variables. The grey prediction model is prone to a rapid increase and decay and a lack of adaptive ability, so it is suitable for short-term prediction. The neural network model needs a large number of training samples. The autoregressive integrated moving-average (ARIMA) model can process linear time series data, but cannot capture nonlinear time series data. The kernel function selection of the support vector machine method is subjective. The combined models do not guarantee the size of the weights. Container throughput has the characteristics of randomness, volatility, and nonlinearity, which increase the difficulty of prediction. Therefore, it is urgent to find a simple, effective, and high-precision method to predict container throughput.

In recent years, the technological progress of computing tools and computational capacity has allowed us to deal with a large number of complex and high-dimensional data, which come from different practical fields. For example, the monthly data of container throughput, the daily data of temperature, the hourly data of electricity quantity, and the minute data of stock prices all have functional characteristics. These data are characterized as curves, shapes, and images, which are usually called functional data. Ref. [1] proposed functional data analysis methods based on functional analysis techniques, the topology, and statistics, which are methods of processing functional data. The main idea is to regard the data observed once in the observation interval as a whole. Functional data form curves, surfaces, and images allow fully understanding the changes of the whole data, finding the change rules, mining the data information, and making data analysis more accurate. At present, functional data analysis has attracted much attention due to its wide application in many practical areas, such as biology, medical sciences, geophysics, meteorology, pattern recognition, and so on.

Container throughput data are a kind of nonlinear time series dataset. In order to achieve the purpose of prediction, the time series methods are very suitable. Time series are an active field in current statistical research because of their wide application in various practical fields. There are many time series methods, such as autoregressive, moving average, grey prediction, ARIMA, seasonal ARIMA models, and so on. However, no research results consider the functional characteristics of container throughput time series data. Recent research has attacked the functional time series problem by proposing either parametric (mainly linear) or nonparametric modeling. In order to overcome the constraint of the linear relationship between variables, in this paper, we propose nonparametric methods with more flexibility and applicability. In 1942, Wolfowitz first used the term nonparametric statistics, which mainly expanded the content of a parametric test so that the traditional test process could be applied to small samples and data of different distribution types. Nonparametric statistics is dedicated to solving the estimation and testing problems of unknown theoretical distributions and dealing with distribution-independent problems. Compared with parametric methods, nonparametric methods have wide applicability and robustness. The nonparametric approaches do not assume specific population distributions; they are applicable to data from any unknown population distribution and can describe more problems. Nonparametric methods have relatively few restrictions on the population distribution.

In order to take a high number of historical data into the time series model, functional data analysis is a reasonable method. The original time series was composed of 15 curves' data, which are the measurements of container throughput in Shanghai Port during 15 years raging from 2007 to 2021. Hence, the time series can be viewed as a list of 12 functional data. We can use a single functional variable to forecast container throughput rather than 12 observed points. Hence, the prediction problem can be addressed through a nonparametric method with dependent functional time series. The aim of this paper was to develop several prediction models that combine the advantages of nonparametric methods

with the merits of the functional idea to achieve the purpose of forecasting the yearly container throughput to compare the prediction accuracy with other classical methods. The results showed that the nonparametric functional data analysis methods outperformed other benchmark models.

## 2. Literature Review

Undoubtedly, an accurate port data analysis can improve a port's operations and is helpful for the economic development of the port's cities and the global economy. The investigations of container throughput forecasting have continued to deepen and provide a theoretical basis for the future development of ports. Classical regression models have traditionally been applied to forecast container throughput and can yield reasonably acceptable forecasting results. For example, Ref. [2] considered the seasonal ARIMA model to forecast the monthly container throughput of the top 20 international container ports, and the results suggested that the seasonal ARIMA model can produce accurate and reliable throughput forecasting at major international ports. Ref. [3] developed the grey model (GM) to forecast the port's throughput, which generated more accurate results. Ref. [4] applied the Fourier series to modify the GM to predict the cargo throughput of Kaohsing Port. Ref. [5] collected the foreign trade data from 1996 to 2008 and constructed a back propagation-artificial neural network (BP-ANN) model to predict container throughput. Ref. [6] proposed a hybrid method by combining projection pursuit regression and the genetic programming algorithm to forecast the container throughput of Qingdao Port. The results showed that the hybrid method significantly outperformed the ANN, seasonal ARIMA, and projection pursuit regression models. Ref. [7] established the GM, triple exponential smoothing model, multiple linear regression model, and back propagation neural network models to forecast the container throughput of both Shanghai Port and Lianyungang Port. Ref. [8] gave a comparison among some time series methods for forecasting container throughput. Ref. [9] showed some univariate models to predict the container throughput of major ports in Asia.

Recently, machine learning approaches have allowed extracting hidden relationships from the real data and have become more and more important for economics data analysis. Many studies have been conducted to improve forecasting accuracy by using machine learning techniques. For example, Ref. [10] proposed the support vector regression (SVR) model to consider potential influencing prediction to forecast container throughput. Ref. [11] conducted several hybrid approaches by combining SVR with other methods to forecast container throughput. The results indicated that a good prediction depends on the seasonal nature and nonlinear characteristics of the historical data. Ref. [12] employed a deep learning method to forecast the container throughput of Los Angeles Port. Ref. [13] provided practical methods for forecasting the container throughout through the random forest and multilayer perception models. The results implied that the random forest model is a reasonable choice by comparison with seven competing models. Ref. [14] considered a long short-term memory (LSTM) deep learning method and seasonal ARIMA model to forecast the comprehensive and route-based Shanghai Containerized Freight Index. The findings showed that the LSTM deep learning model outperformed the seasonal ARIMA model.

In the past two decades, functional data analysis has received considerable attention due to expressing each individual datum in repeatedly measured data as a smooth and continuous time processes and drawing information from functional data. There are a significant amount of works devoted to functional data analysis, especially in the fields of medicine, the environment, sports, economics, and so on. For example, Ref. [15] treated the data related to the intraocular pressure of patients with right eye glaucoma by functional analysis and prediction based on smoothed functional principal component analysis (FPCA). Ref. [16] proposed the ranking of functional data by several ranking methods and applied worldwide PM10 data to generate ranks. Ref. [17] employed regularized optimization and functional principal component analysisfor prediction and estimation of sparse functional data and forecasting the behavior of basketball players. Ref. [18]

introduced several ordinal classification methods for multiargument and multivariate functional data. Their performance was analyzed on four real datasets of the 3D brain structure. Ref. [19] explored the problems of detecting outliers in German electricity supply functions and classification of medical imaging data by partially observed functional data with an integrated functional depth.

Much statistical literature has focused on functional time series by proposing parametric and nonparametric modeling. Ref. [20] considered the estimation and prediction of linear processes in the functional space. In order to reduce modeling biases caused by the misspecifying parametric models, there has been an upsurge in interest and effort in nonparametric modeling, which is helpful to explore hidden structures and reducing modeling biases. Nonparametric functional models have been investigated intensively in real data analysis. For example, Ref. [21] studied the theoretical results of some kernel estimators of the model and how the procedure was well adapted to some spectrometric data. Ref. [22] used the reproducing kernel Hilbert spaces framework to consider the functional nonparametric regression model. Ref. [23] considered Gaussian processes methods for a nonparametric functional regression model and applied Leeds renal anemia data to confirm the usefulness of the method. The results showed that the proposed method outperformed the kernel method, functional additive model, penalized function-on-function regression, and functional linear model.

In order to import a large number of past values into a nonparametric time series model by applying the functional data idea, we need to cut the observed time series into one sample of trajectories and bring in the model one single past and continuous trajectory rather than many single past values. The literature on functional time series is quite extensive. For instance, Ref. [24] estimated the conditional mode by maximizing a kernel estimator of conditional density for the response and almost achieved the convergence of the proposed estimator under $\alpha$-mixing assumption. Ref. [25] considered a semi-functional partial linear model for time series prediction. Ref. [26] established the almost complete convergence and asymptotic normality of a nonparametric conditional mode estimator and illustrated the proposed method by using functional El Niño time series data. Ref. [27] used a nonparametric learning approach based on the support vector machines technique to estimate functional quantiles and applied it to El Niño time series prediction. Ref. [28] investigated $k$-nearest-neighbor (kNN) estimation for a strong mixing functional time series model and established the uniform almost complete convergence rate of the proposed estimator. The finite sample performance and the usefulness of the kNN method were illustrated by an empirical application to a real data analysis of sea surface temperature. Ref. [29] discussed kernel regression estimation for time series data with functional response and covariates. Ref. [30] addressed the problem of nonparametric trend estimation for functional time series and applied the proposed method to annual mortality rates in France.

By summarizing the literature on forecasting container throughput, it can be found that the current research works are mainly concentrated on discrete time series and a linear assumption. There are few analyses of the functional characteristics of the time series dataset and a flexible structure. Therefore, the main contributions of this paper include two aspects: (1) We consider functional data analysis methods for the container throughput of Shanghai Port as continuous trajectoryof a single point, which finds more information from the functional time series dataset and produces more accurate predictions than nonfunctional methods. (2) Our research establishes a more flexible relationship between variables via nonparametric approaches to eliminate the linear constraint, which yields more accurate predictions and more extensive applications than parametric methods. Thanks to the advantages of functional data and nonparametric methods, we propose nonparametric functional data analysis approaches, namely the functional regression model, functional conditional quantiles model, and functional conditional mode model, to forecast the container throughput time series.

The remainder of this paper is organized as follows. We describe and preprocess the container throughput time series data of Shanghai Port from 2007 to 2021 in Section 3. Three nonparametric functional data analysis methods and the corresponding kernel estimations are presented in Section 4. Section 5 compares the prediction accuracy of the nonparametric functional models with other benchmark models. Section 6 gives the concluding comments and future research ideas.

### 3. Data

Shanghai Port is located in Shanghai, China; see Figure 1 for the details. Shanghai Port gives full play to its advantages, supports the economic development of Shanghai, and realizes the positive interaction between the port and its cities. The container throughput of Shanghai Port reached 47 million TEUs on 1 January 2022, ranking first in the world for 12 consecutive years. This section focuses on an economic time series datasets. The Shanghai Port monthly container throughput covering a period from January 2007 to December 2021 (180 months) is plotted in Figure 2. It can be seen that the container throughput has been seriously affected and fell sharply at the beginning of 2020 due to the outbreak of COVID-19.

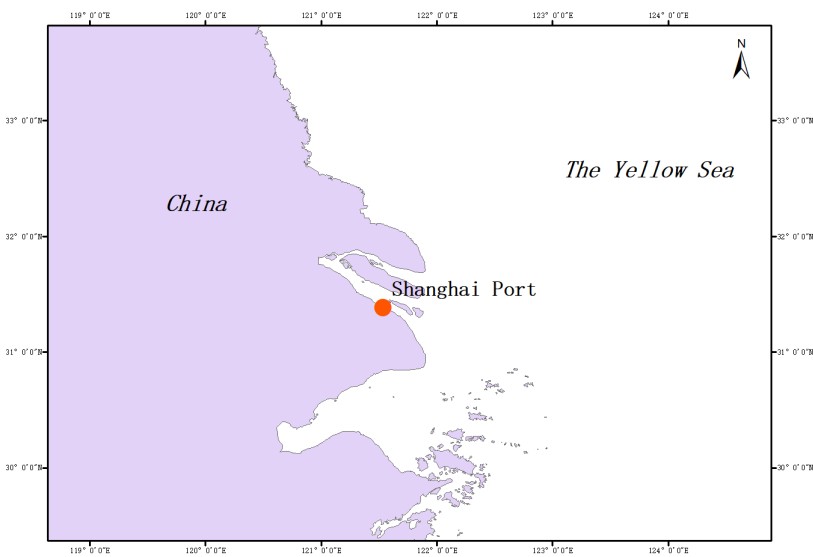

**Figure 1.** The location of Shanghai Port in China.

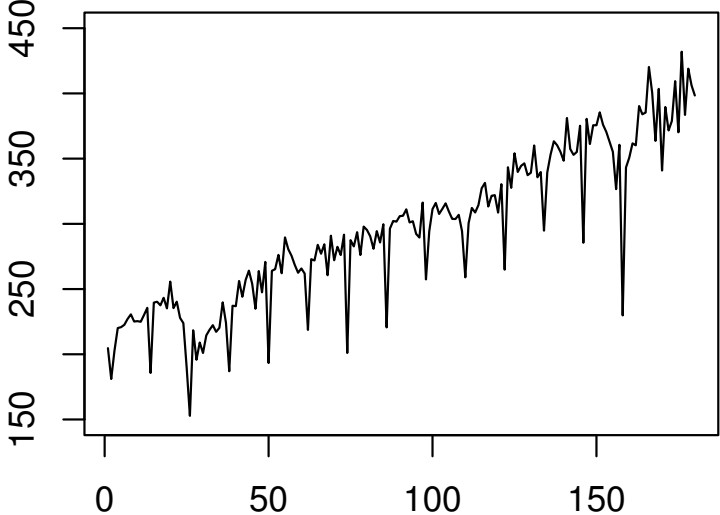

**Figure 2.** The monthly container throughput of Shanghai Port.

It is obviously seen that the time series exhibits a linear trend and heterogeneity in the variance structure from Figure 2. In order to eliminate these effects, we took a new time series by differentiating the log data. The transformed time series is displayed in Figure 3.

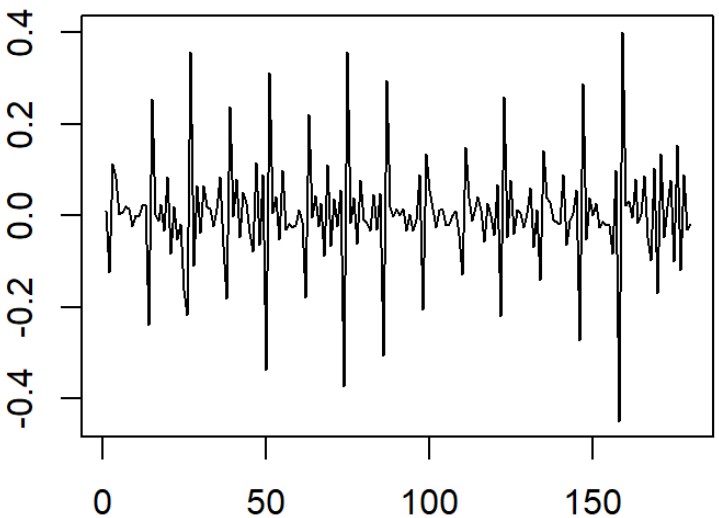

**Figure 3.** The monthly differentiated log container throughput of Shanghai Port.

In order to predict the future container throughput, the classical statistical methods take a finite number of historical data into account. It is reasonable to consider the continuous time series over some period as the explanatory variables. Hence, the explanatory variables were composed of 15 yearly continuous time series. The functional data are given in Figure 4.

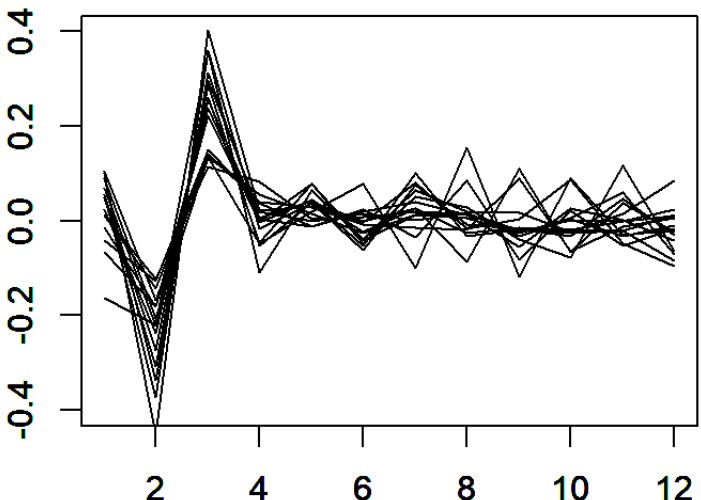

**Figure 4.** The yearly differentiated log container throughput of Shanghai Port.

The first eigenspace both of the variables and units is given in Figure 5. It is clear that the structure is visible, which confirms that the historical time series cannot be summarized by a small number of parameters. That is to say, it would be accurate if the whole of past years were used to predict the future values.

The descriptive statistic results are listed in Table 1. Because the range of the original data is huge and has a clear linear trend, for the ease of calculation and correcting the non-normal distribution problem, as well as effectively eliminating the trend, we reduced the absolute value of the original data by using the natural differentiated logarithmic value.

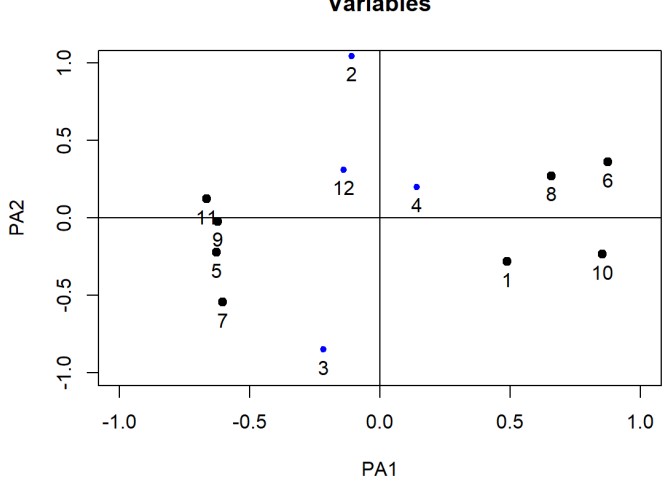

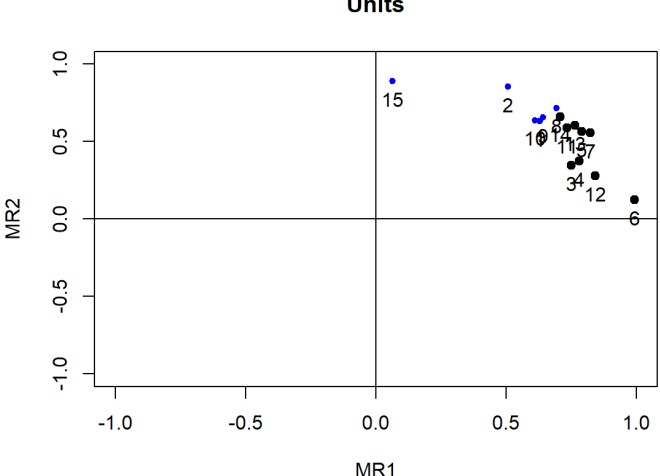

**Figure 5.** The standard FPCA of the container throughput of Shanghai Port.

**Table 1.** The descriptive statistic results of container throughput in Shanghai Port.

| Data | Mean | Max | Min | Std. Dev. | Skewness | Kurtosis | J-B Test |
|---|---|---|---|---|---|---|---|
| Original data | 295.17 | 432 | 152.8 | 59.68 | 0.08 | −0.81 | 109.06 *** |
| Log data | 5.67 | 6.07 | 5.03 | 0.21 | −0.31 | −0.56 | 95.05 *** |
| Differentiated log data | 0 | 0.4 | −0.45 | 0.12 | −0.05 | 3.42 | 1.398 *** |

Notes: (1) J-B test is the Jarque–Bera normal distribution test statistic. (2) *** stands for the statistical value being significant at 0.01.

The functional approach to time series forecasting consists of taking the historical explanatory data as a whole continuous path of the process. Without loss of generality, we assumed that $N = n\tau$ for some $n \in \mathbb{N}^*$ and $\tau > 0$. To clarify this, in the container throughput, we have $N = 180$, $n = 15$, and $\tau = 12$. The observed time series can be cut into $n$ continuous paths of length $\tau$. Hence, we can build a new sample size of $n - 1$ in the following way:

$$\mathcal{X}_i = \{Z_t, (i-1)\tau < t \leq i\tau\} \text{ and } Y_i = Z(i\tau + s), i = 1, \cdots, n-1, \tag{1}$$

where $\{Z_t, t \in [0, +\infty)\}$ is a real-valued time series, which has been observed $N$ equispaced times. $s$ is the horizon of prediction. The forecasting problem turns into a prediction question of the scalar response $Y_i$ given functional variable $\mathcal{X}_i$.

## 4. Methodology

The functional approach to time series can be extended to a more general response values of following form:

$$\mathcal{X}_i = \{Z_t, (i-1)\tau < t \leq i\tau\} \text{ and } Y_i = g(\mathcal{X}_{i+1}), \ i = 1, \cdots, n-1, \tag{2}$$

where $g(\cdot)$ is a known function with a real value. The forecasting question turns into predicting some characteristic of the future response $Y_i = g(\mathcal{X}_{i+1})$ for some fixed and real value $g(\cdot)$ from the information obtained in the last period $\{\mathcal{X}_i, i = 1, \cdots, n-1\}$. In this paper, we developed standard tools such as nonparametric functional regression and functional conditional quantile and functional conditional mode models to analyze the container throughput time series data, which were summarized in [31].

### 4.1. Nonparametric Functional Regression Model

Assume that a sample of identically distributed pairs $\{(\mathcal{X}_i, Y_i), \ i = 1, \cdots, n\}$. The functional nonparametric functional regression is defined as follows:

$$Y_i = \gamma(\mathcal{X}_i) + \varepsilon_i, \ i = 1, \cdots, n, \tag{3}$$

where $\mathcal{X}_i$ is functional covariate defined on the interval $I$. $Y_i$ is a real scalar response. The observed explanatory variable $\mathcal{X}_i$ is a functional covariable value in a semi-metric $d(\cdot, \cdot)$. $\gamma(\cdot)$ is an nonlinear operator. $\varepsilon_i$ is an error random variable with $E(\varepsilon_i | \mathcal{X}_i) = 0$ and $Var(\varepsilon_i) = \sigma^2$. The model has wide applicability and flexibility and is free from the linear constraints. In addition, the functional linear model $Y = \int_I \mathcal{X}(t)\beta(t)dt + \varepsilon$ is a special case of the model.

In application practice, the operator $\gamma(\cdot)$ is unknown. In order to estimate the nonlinear operator $\gamma(\cdot)$, the Nadaraya–Watson kernel method was introduced to estimate $\gamma(\cdot)$, which is defined by

$$\widehat{\gamma}_n(\mathcal{X}) = \frac{\sum_{i=1}^n Y_i K(h^{-1}d(\mathcal{X}, \mathcal{X}_i))}{\sum_{i=1}^n K(h^{-1}d(\mathcal{X}, \mathcal{X}_i))},$$

where $K$ is an asymmetrical kernel function and $0 < h = h_n \to 0$ is a sequence of smoothing parameters as $n \to \infty$.

### 4.2. Functional Condition Quantiles Model

In this subsection, we attack the prediction problem of the scalar response $Y$ given the functional predictor $\mathcal{X}$ by functional conditional quantiles. The nonlinear conditional cumulative distribution operator is defined by

$$F(\mathcal{X}, y) = P(Y \leq y | \mathcal{X}), \ y \in R.$$

Hence, for $\alpha \in (0, 1)$, the functional conditional quantile is defined by

$$t_\alpha(\mathcal{X}) = \inf\{F(\mathcal{X}, y) \geq \alpha\}, \ y \in R. \tag{4}$$

We can define a kernel estimator of the functional conditional quantiles $t_\alpha(\mathcal{X})$ as follows:

$$\widehat{t}_\alpha(\mathcal{X}) = \inf\{\widehat{F}(\mathcal{X}, y) \geq \alpha\}, \ y \in R$$

where the kernel estimator of $F(\cdot, \cdot)$ is given by

$$\widehat{F}(\mathcal{X}_i, y) = \frac{\sum_{i=1}^n Y_i K(h^{-1}d(\mathcal{X}, \mathcal{X}_i)) H(b^{-1}(y - Y_i))}{\sum_{i=1}^n K(h^{-1}d(\mathcal{X}, \mathcal{X}_i))}$$

with an integrated kernel function $H$ and a sequence of smoothing parameters $0 < b = b_n \to 0$.

### 4.3. Functional Conditional Mode Model

In this subsection, we attack the prediction problem of the scalar response $Y$ given the functional predictor $\mathcal{X}$ by the functional conditional mode. Under differentiability assumption, the nonlinear conditional density operator is defined by

$$f(\mathcal{X}, y) = \frac{\partial}{\partial y} F(\mathcal{X}, y).$$

The functional conditional mode is defined as follows:

$$\theta(\mathcal{X}) = \arg \sup_{y \in S} f(\mathcal{X}, y), \ S \subset R. \tag{5}$$

The kernel estimators of $f(\cdot, \cdot)$ and $\theta(\cdot)$ are, respectively, defined by

$$\widehat{f}(\mathcal{X}, y) = \frac{b^{-1} \sum_{i=1}^{n} Y_i K(h^{-1} d(\mathcal{X}, \mathcal{X}_i)) G(b^{-1}(y - Y_i))}{\sum_{i=1}^{n} K(h^{-1} d(\mathcal{X}, \mathcal{X}_i))}$$

and

$$\widehat{\theta}(\mathcal{X}) = \arg \sup_{y \in S} \widehat{f}(\mathcal{X}, y), \ S \subset R,$$

where $G$ is another kernel function.

## 5. Empirical Analysis

### 5.1. Data Description

This section focuses on an application to economic time series coming from the container throughput of Shanghai Port described in Figure 2. We downloaded the monthly container throughput covering a period from January 2007 to December 2021 (180 months) from the official web page (https://www.portshanghai.com.cn/tjsj/index.jhtml) (accessed on 15 January 2022) of Shanghai International Port Group. The differentiated log data were recorded as a sequence of real numbers, which are composed of $N = 180$ real data. To better use the functional methodology, we cut the original time series into a set of functional data. Hence, we chose $\tau = 12$. The data were put into a new matrix file of size $15 \times 12$, which is organized in Table 2. For the purpose of prediction, we used the means of the data corresponding to the 14 previous data to predict the 15th year.

**Table 2.** The yearly container throughput expressed by a functional dataset.

| Number | Month 1 | Month 2 | $\cdots$ | Month $j$ | $\cdots$ | Month 11 | Month 12 |
|---|---|---|---|---|---|---|---|
| Year 1 | $z_1$ | $z_2$ | $\cdots$ | $z_j$ | $\cdots$ | $z_{11}$ | $z_{12}$ |
| Year 2 | $z_{13}$ | $z_{14}$ | $\cdots$ | $z_j$ | $\cdots$ | $z_{23}$ | $z_{24}$ |
| $\vdots$ | | | | | | | |
| Year $i$ | $z_{1+12(i-1)}$ | $z_{2+12(i-1)}$ | $\cdots$ | $z_{j+12(i-1)}$ | $\cdots$ | $z_{11+12(i-1)}$ | $z_{12+12(i-1)}$ |
| $\vdots$ | | | | | | | |
| Year 14 | $z_{217}$ | $z_{158}$ | $\cdots$ | $z_{j+156}$ | $\cdots$ | $z_{167}$ | $z_{168}$ |
| Year 15 | $z_{169}$ | $z_{170}$ | $\cdots$ | $z_{j+12(i-1)}$ | $\cdots$ | $z_{179}$ | $z_{180}$ |

### 5.2. Evaluation Criteria

In order to measure the performance of each prediction method, we considered the following three evaluation criteria, i.e., mean-squared error, mean average error, and mean average percent error. Given 12 pairs of the observed value $y_i$ and the predicted value $\widehat{y}_i$, the three evaluation criteria are the measure of accuracy of $\widehat{y}_i$ simulating $y_i$ and are defined

as follows:

(1) Mean-squared error (MSE):

$$\text{MSE} = \frac{1}{12} \sum_{i=1}^{12} (y_i - \widehat{y}_i)^2, i = 1, \cdots, 12.$$

(2) Mean average error (MAE):

$$\text{MAE} = \frac{1}{12} \sum_{i=1}^{12} |y_i - \widehat{y}_i|, i = 1, \cdots, 12.$$

(3) Mean average percent error (MAPE):

$$\text{MAPE} = \frac{1}{12} \sum_{i=1}^{12} \frac{|y_i - \widehat{y}_i|}{y_i}, i = 1, \cdots, 12.$$

*5.3. Parameters Setting*

Before the simulation analysis, some parameters need to be determined. In the non-parametric functional model, the semi-metric, kernel functions, and bandwidths have to be predefined. In terms of the smoothness of the simulation curves, we used the FPCA semi-metric, which has the following form

$$d_q^{FPCA}(\mathcal{X}, \mathcal{X}_i) = \left\{ \sum_{i=1}^{q} \left( \int [\mathcal{X}(t) - \mathcal{X}_i(t)] v_k(t) dt \right)^2 \right\}^{1/2}, k = 1, 2, \cdots,$$

where $v_1, v_2, \cdots$ are the orthonormal eigenfunctions of the covariance operator $E[\mathcal{X}(t)\mathcal{X}(s)]$. $q$ is a tuning parameter, which allows us to obtain the best empirical mean-squared errors; here, we took $q = 2$. In following simulations, we took the kernel functions to be the quadratic kernel, which is defined as follows:

$$K(x) = H(x) = G(x) = \frac{3}{4}(1 - x^2) I_{[0,1]}(x).$$

In addition, motivated in Ferraty and Vieu (2002), the optimal bandwidths $h_{opt}$ and $b_{opt}$ were obtained by the cross-validation procedure.

*5.4. Comparison of Different Models*

To demonstrate the usefulness of the proposed models, the statistical models used to compare the different predictions are given in Table 3. We first compared the FNR model against NR model to illustrate the advantages of the functional data. Subsequently, we compared the FNR model to the FLR model to show that the non-parametric method is superior to the parametric method. After that, to demonstrate the overall advantage of the functional time series over classical time series, the FNR, FCQ, and FCM models were compared to the ARIMA model, because the monthly container throughput data show regular fluctuations. Finally, in order to select the best nonparametric functional method, we further compared the performance of the FNR, FCR, and FCM models. In this simulation, we applied the MSE, MAE, and MAPE to measure the accuracy of the forecasting models. The comparison results are presented in Tables 4–6 using the R software.

**Table 3.** The different prediction models.

| Models | Definition | Notations |
|---|---|---|
| Functional nonparametric regression model | $Y = \gamma(\mathcal{X}) + \varepsilon$ | FNR |
| Functional conditional quantile | $t_\alpha(\mathcal{X}) = \inf\{F(\mathcal{X}, y) \geq \alpha\}$ | FCQ |
| Functional conditional mode | $\theta(\mathcal{X}) = \arg\sup_{y \in S} f(\mathcal{X}, y)$ | FCM |
| Functional linear regression model | $Y = \int \mathcal{X}(t)\beta(t)dt + \varepsilon$ | FLR |
| Nonparametric regression model | $Y = \gamma(X) + \varepsilon$ | NR |
| Classical time series models | $\text{ARIMA}(0, 0, 1)$ | ARIMA |

(1) Comparison between different nonparametric regression models:

Figure 6 describes the curves of the FNR and NR models. Table 4 compares their performance in terms of the MSE, MAE, and MAPE. From Figure 6 and Table 4, we can draw the conclusion that the FNR model had smaller errors than the NR model, which implies that functional data produce more accurate predictions. The reason may be that the classical nonparametric regression model treats the original time series as single points regardless of the continuity of the dataset.

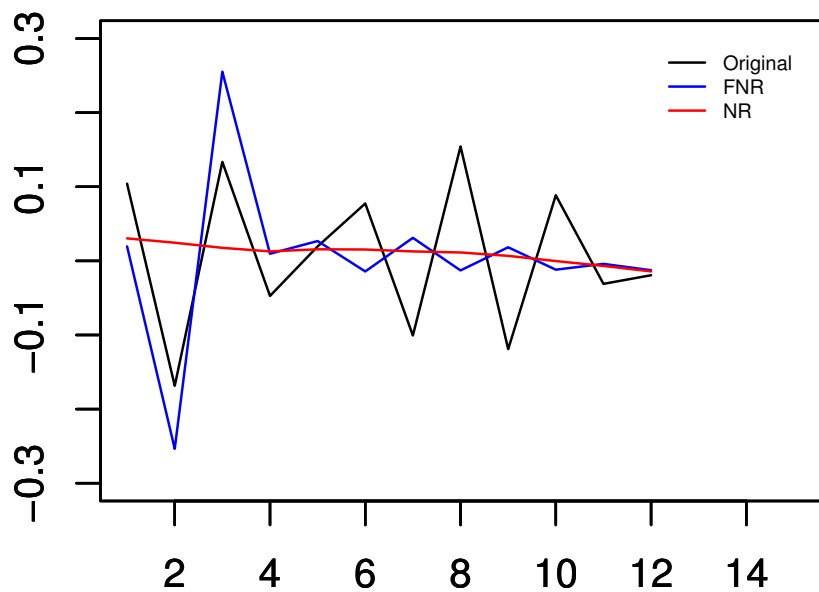

**Figure 6.** The curves of the true value, FNR model and NR model.

**Table 4.** The results of forecasting errors for different models.

| Models | MSE | MAE | MAPE |
|---|---|---|---|
| FNR | 0.0097 | 0.0848 | 0.9076 |
| NR | 0.0110 | 0.0896 | 0.9609 |

(2) Comparison between different functional regression models:

Figure 7 describes the curves of the FNR and FLR models. Table 5 compares their performance in terms of prediction errors. From Figure 7 and Table 5, it can be seen that the FNR model achieved smaller errors than the FLR model, which implies that the nonparametric estimation method yields more accurate predictions. The cause might be that the FLR model assumes that the relationship between historical data and current data is linear, ignoring the nonlinear fluctuations of the time series dataset.

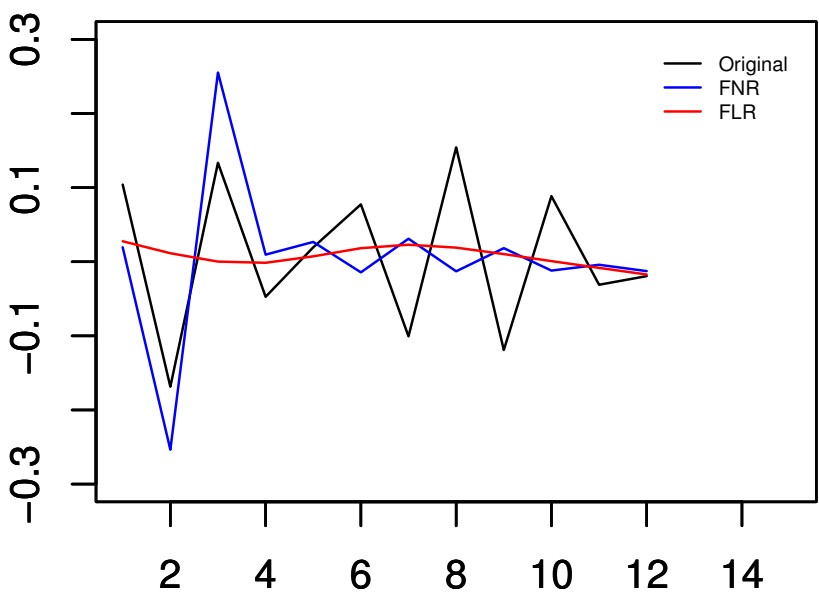

**Figure 7.** The curves of the true value, FNR model and FLR model.

**Table 5.** The results of forecasting errors for different models.

| Models | MSE | MAE | MAPE |
|---|---|---|---|
| FNR | 0.0097 | 0.0848 | 0.9076 |
| FLR | 0.0109 | 0.0887 | 1.0793 |

(3) Comparison among different time series models:

In this simulation, we selected the optimal model ARIMA(0,0,1) by the AIC and BIC. In Figure 8, plots of the three functional nonparametric forecasting methods and the ARIMA model are presented. Table 6 gives the prediction errors to compare their performance. Figure 8 and Table 6 show that the functional nonparametric methods for time series forecasting were more accurate than the classical ARIMA model in the same setting. Essentially, the ARIMA model can only capture the linear relationship, but not the nonlinear relationship. In addition, the FCQ model performed better than the FNR and FCM models.

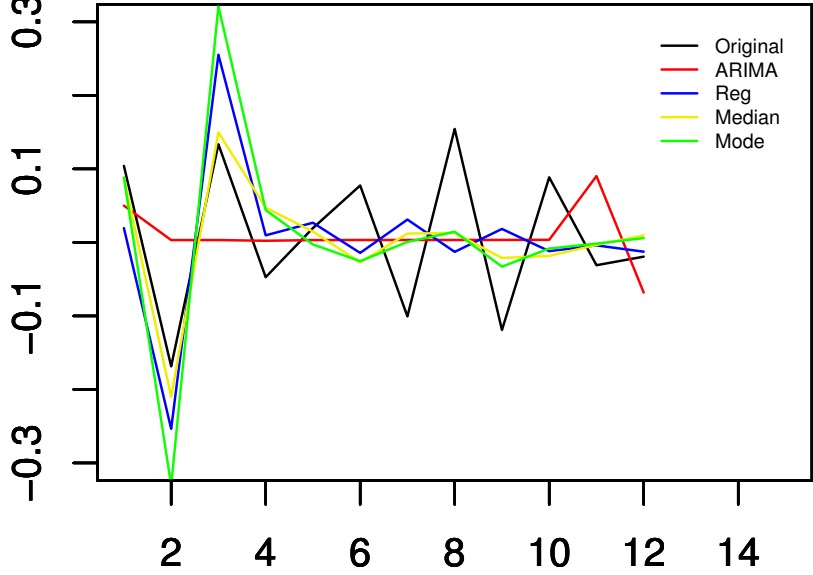

**Figure 8.** The curves of the true value, FNR model, FCQ model, FCM model, and ARIMA model.

**Table 6.** The results of forecasting errors for different models.

| Models | MSE | MAE | MAPE |
| --- | --- | --- | --- |
| FNR | 0.0097 | 0.0848 | 0.9076 |
| FCQ | 0.0065 | 0.00661 | 0.8795 |
| FCM | 0.0108 | 0.0884 | 1.0751 |
| ARIMA | 0.0109 | 0.0941 | 1.3151 |

(4) Comparison among different functional nonparametric models:

In Figures 9–11, each of the three plots is concerned with the different forecasting methods (FNR, FCQ, or FCM). What can be said from these results is that each of the three functional approaches for time series forecasting gave appealing results on this dataset. Specially, the FCQ model's performance had a slight advantage over the FNR and FCM models.

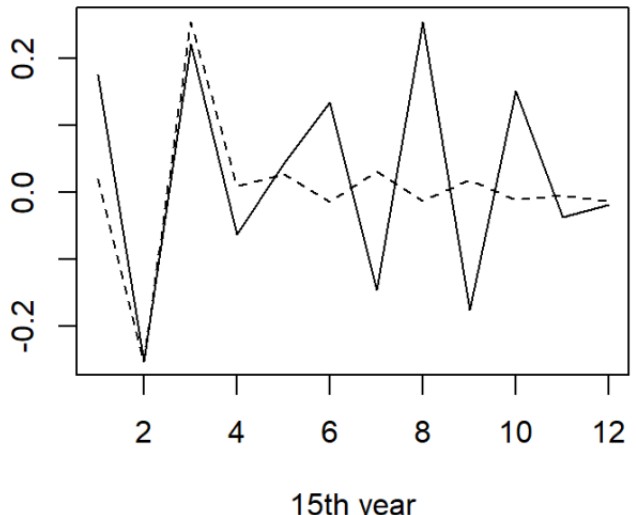

**Figure 9.** The curves of the true value (solid line) and FNR model (dotted line) with MSE = 0.0097.

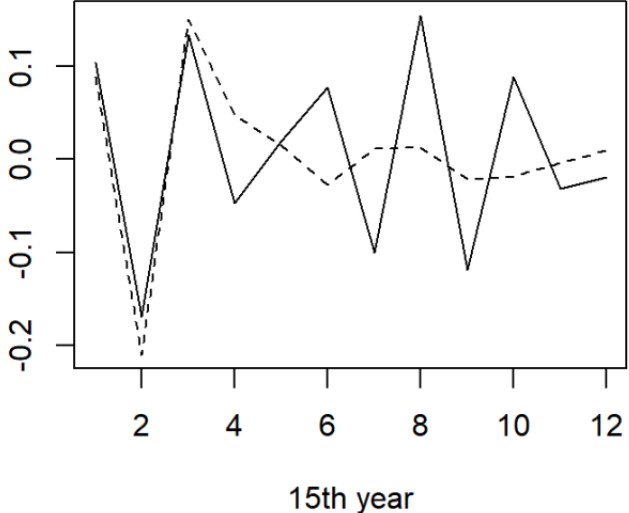

**Figure 10.** The curves of the true value (solid line) and FCQ model (dotted line) with MSE = 0.0065.

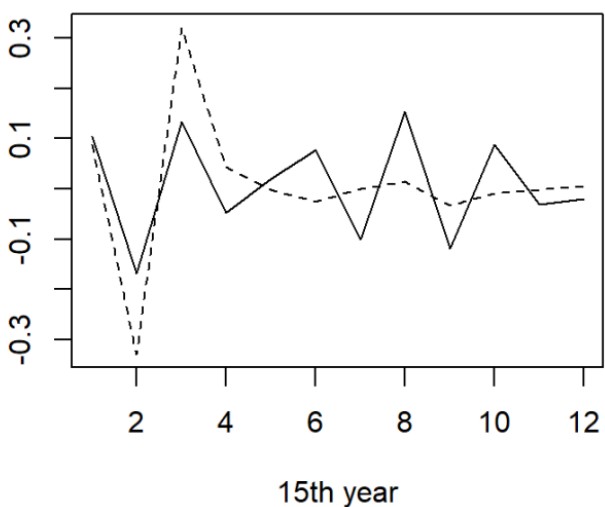

**Figure 11.** The curves of the true value (solid line) and FCM model (dotted line) with MSE = 0.0108.

## 6. Concluding Remarks

Container throughput time series data show strong functional characteristics with the passage of time. However, in the existing literature, only single past values ignore the continuity of the past values. In this research, we developed one new statistical method, nonparametric functional data analysis, to predict container throughput time series by taking into account one continuous set of past values as the predictors, which not only effectively avoided the limitation of the linearity assumption, but also could mine more data information and produce more accurate predictions.

A comparison among the FNR, FCQ, FCM, NR, FLR, and ARIMA models was considered and applied to forecast the container throughput time series of Shanghai Port. The models were proposed to forecast the container throughput of the 15th year, for which the historical dataset in a 14 year period is available. The forecasting accuracy results of the proposed models were presented in the form of the measurement criteria MSE, MAE, and MAPE. At the same time, the forecasting curves of all the prediction models were plotted. The conclusions suggested that: (1) functional data find more information than discrete points; (2) nonparametric methods are more widely applied and flexible than parametric methods; (3) functional time series analysis outperforms classical time series models. On the other hand, among these three nonparametric functional forecasting models, the best forecasting performance could be obtained by means of the FCQ model.

Due to the lack of scientific works focusing on forecasting container throughput by functional data analysis, the main contribution of this paper shows that the nonparametric functional models are reliable for forecasting container throughput. More accurate forecasting results help with management decisions.

It is interesting and challenging to study the main factors affecting the comprehensive level of ports by functional principal component analysis method, and classify ports based on the functional cluster analysis method, which will be used to compare with the traditional principal component analysis and cluster analysis. The goal is to classify ports more effectively and optimize port indicators more accurately.

**Author Contributions:** Conceptualization, Y.Z.; methodology, Y.Z.; software, B.S.; writing—original draft preparation, Y.Z.; writing—review and editing, Y.C. All authors have read and agreed to the published version of the manuscript.

**Funding:** This work is supported by National Natural Science Foundation of China (12101393) and the National Statistical Science Research Project (2021LZ41).

**Institutional Review Board Statement:** Not applicable.

**Informed Consent Statement:** Not applicable.

**Data Availability Statement:** Readers can access our data by sending an email to the corresponding author Yuye Zou.

**Acknowledgments:** The authors thank the Editors and Referees for their carefully reading of this manuscript and for their helpful comments and suggests.

**Conflicts of Interest:** The authors declare no conflict of interest.

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
