# Peer review of "Nonparametric Functional Data Analysis for Forecasting Container Throughput: The Case of Shanghai Port"

_jmse, doi:10.3390/jmse10111712_

Round 1

Reviewer 1 Report

The methodology used in the paper is relatively simple but it fulfills its task. Using 12 months as a period of data seems obvious but should be tested for any possible anomalies.

One correction required is the fact that there is no direct source of the data. It should be clearly stated where they came from, when they were acquired and relevant web page ought to be referenced in bibliography.

Author Response

From the perspective of economic statistics, monthly data are cyclical. In addition to the chain relative ratio, the year-on-year change is also the focus of our research. This is also consistent with the statistical data release cycle.

In June this year, the authors downloaded the monthly container throughput covering a period from January 2007 to December 2021 (180 months) from the official website (https://www.portshanghai.com.cn/tjsj/index.jhtml) of Shanghai International Port Group. Section 5 of the manuscript has described the source and web page of the data.

Reviewer 2 Report

First, congratulations for the excellent work. Below I will insert some considerations about the work.

1. I missed references in the introduction to restate your point of view or problem in question. 

2. I think the introduction is too long. If you hadn't done the literature review section, it could be like this, but as you did, I believe that a lot of the things you described in the introduction could be left in the literature review section. So try to make the introduction section leaner.

3. Paragraphs that are too long, making reading tiring, try to divide your paragraphs well.

4. You make an applied comparison of your proposed model with classical methods, but I don't see a discussion of the model proposed with the studies of the literature review.

5. Implications of the study are missing in the manuscript.

6. Leave directly in the text, the implications of future work perspectives.

Best regards. 

Author Response

The authors have replied to the reviewer's comments one by one in the attachment.
